Extended Abstract Track

# Learning Deep Generative Models with Invariance under Symmetry Transformations

**James Urquhart Allingham**                                    JUA23@CAM.AC.UK

**Javier Antorán**                                            JA666@CAM.AC.UK

**Shreyas Padhy**                                             SP2058@CAM.AC.UK

**Eric Nalisnick**[†]                                         E.T.NALISNICK@UVA.NL

**José Miguel Hernández-Lobato**                              JMH233@CAM.AC.UK

*University of Cambridge,* [†] *University of Amsterdam*

**Editors:** Sophia Sanborn, Christian Shewmake, Simone Azeglio, Arianna Di Bernardo, Nina Miolane

## Abstract

While imbuing a model with invariance under symmetry transformations can improve data efficiency and predictive performance, most methods require specialised architectures and, thus, prior knowledge of the symmetries. Unfortunately, we don't always know what symmetries are present in the data. Recent work has solved this problem by jointly learning the invariance (or the degree of invariance) with the model from the data alone. But, this work has focused on discriminative models. We describe a method for learning invariant *generative* models. We demonstrate that our method can learn a generative model of handwritten digits that is invariant to rotation.

**Keywords:** deep generative models, learned invariance, symmetry.

## 1. Introduction

Generative models are often motivated as being more data efficient than discriminative models (Welling, 2019). We expect that since we are encoding our beliefs about how the data are generated (our inductive biases) into the model, it will be able to learn from fewer data. Unfortunately, deep generative models are *more* data hungry than their discriminative counterparts. This is because deep generative models tend to only incorporate very general inductive biases (smoothness, hierarchies of random variables, etc.). One kind of inductive bias that does improve efficiency is invariance under symmetry transformations in the data (van der Ouderaa and van der Wilk, 2022). In some cases, we know apriori which transformations the model should be invariant to. However, in many cases, we need to learn when (or to what extent) to be invariant. For example, in the case of handwritten digit recognition, we know that we should be invariant to some amount of rotation. However, a model that is invariant to rotations in the full range $[-180°, 180°]$ would be unable to distinguish between '6' and '9'. Thus, the degree of invariance should be learned. Recent work has focused on solving this problem for discriminative models (Nalisnick and Smyth, 2018; van der Wilk et al., 2018; Benton et al., 2020; Schwöbel et al., 2021; van der Ouderaa and van der Wilk, 2022; Immer et al., 2022). But, the problem has been neglected for deep generative models – our focus in this paper. We aim to construct a deep generative model $p(\mathbf{x})$ which is invariant to symmetry transformations of $\mathbf{x}$. The generative model

ALLINGHAM ANTORÁN PADHY NALISNICK[†] HERNÁNDEZ-LOBATO

should be equipped with invariant representations $\hat{\mathbf{x}}$ containing no information about the transformation. Concretely, $p(\hat{\mathbf{x}} \,|\, \mathbf{x}) = p(\hat{\mathbf{x}} \,|\, \mathcal{T}_{\boldsymbol{\eta}}(\mathbf{x}))$, where $\boldsymbol{\eta}$ is a random variable which parameterizes the transformation $\mathcal{T}$. For example, $\mathcal{T}$ could be the rotation of an image, and $\eta$ could be the angle. The degree of invariance learnt—i.e., $\boldsymbol{\eta}$—should match the true invariance of the data. That is, unlike Benton et al. (2020) we should not learn a 'maximum data augmentation'.

## 2. The Model and Design Choices

We assume the following generative model for the data $\mathbf{x}$, also depicted in Figure 1:

$$\hat{\mathbf{x}} \sim p_{\boldsymbol{\theta}}(\hat{\mathbf{x}}), \tag{1}$$

$$\boldsymbol{\eta} \sim p_{\boldsymbol{\psi}}(\boldsymbol{\eta} \,|\, \hat{\mathbf{x}}), \tag{2}$$

$$\mathbf{x} \sim p(\mathbf{x} \,|\, \boldsymbol{\eta}, \, \hat{\mathbf{x}}), \tag{3}$$

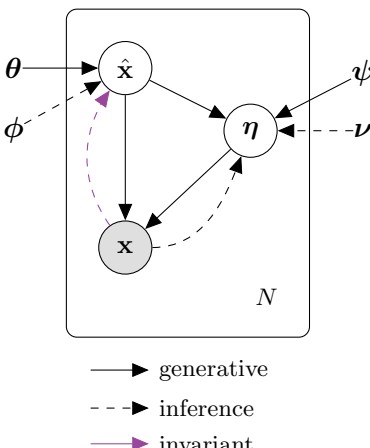

where the *prototype* $\hat{\mathbf{x}}$ can be considered a standardised or reference example with no transformation applied to it. For example, this could be a '6' at in the upright $(90°)$ rotation[1]. Here, $p_{\boldsymbol{\theta}}(\hat{\mathbf{x}})$ is some generative model with parameters $\boldsymbol{\theta}$— a Variational Autoencoder (VAE) or a normalizing flow, for example. Similarly, $p_{\boldsymbol{\psi}}(\boldsymbol{\eta} \,|\, \hat{\mathbf{x}})$ is a neural network (NN),

Figure 1: Graphical model.

with parameters $\boldsymbol{\psi}$, that predicts the distribution over transformations for a given prototype. We choose $p(\mathbf{x} \,|\, \hat{\mathbf{x}}, \, \boldsymbol{\eta}) = \mathcal{N}(\mathcal{T}_{\boldsymbol{\eta}}(\hat{\mathbf{x}}), \sigma \mathbb{I})$. Note that while we do need to specify the kinds of symmetry transformations $\mathcal{T}$ we expect to see in the data, by learning $\boldsymbol{\eta}$ the model will control the degree to which it is invariant. Thus, we can specify several potential symmetry transformations, and learn not to be invariant to those not actually present in the data. E.g., we could specify $\mathcal{T}_{\boldsymbol{\eta}}$ as a family of affine transformation, as in Benton et al. (2020):

$$\mathcal{T}_{\boldsymbol{\eta}}(\hat{\mathbf{x}}) = T_{\boldsymbol{\eta}} \cdot \hat{\mathbf{x}}, \quad T_{\boldsymbol{\eta}} = \exp\left(\sum_i \eta_i G_i\right), \tag{4}$$

where $G_i$ are infinitesimal generator matrices for various affine transformations. For all transformations without symmetries in the data, we expect to learn $p(\eta_i \,|\, \hat{\mathbf{x}}) \approx \delta(\eta_i - 0)$.

We propose a VAE-like inference strategy for this model. Given an inference model—depicted in Figure 1—we can derive an Evidence Lower BOund (ELBO) for jointly learning the generative $\{\boldsymbol{\theta}, \, \boldsymbol{\psi}\}$ and inference $\{\boldsymbol{\phi}, \, \boldsymbol{\nu}\}$ parameters, with gradient descent:

$$\log p(\mathbf{x}) = \log \iint p(\mathbf{x}, \, \hat{\mathbf{x}}, \, \boldsymbol{\eta}) \, d\boldsymbol{\eta} \, d\hat{\mathbf{x}} \tag{5}$$

$$= \log \mathop{\mathbb{E}}_{q_{\boldsymbol{\nu}}(\boldsymbol{\eta} \,|\, \mathbf{x}) \, q_{\boldsymbol{\phi}}(\hat{\mathbf{x}} \,|\, \mathbf{x})} \left[ \frac{p(\mathbf{x} \,|\, \hat{\mathbf{x}}, \, \boldsymbol{\eta}) \, p_{\boldsymbol{\psi}}(\boldsymbol{\eta} \,|\, \hat{\mathbf{x}}) \, p_{\boldsymbol{\theta}}(\hat{\mathbf{x}})}{q_{\boldsymbol{\nu}}(\boldsymbol{\eta} \,|\, \mathbf{x}) \, q_{\boldsymbol{\phi}}(\hat{\mathbf{x}} \,|\, \mathbf{x})} \right] \tag{6}$$

$$\geq \mathop{\mathbb{E}}_{q_{\boldsymbol{\nu}} \, q_{\boldsymbol{\phi}}} \left[ \log p(\mathbf{x} \,|\, \hat{\mathbf{x}}, \, \boldsymbol{\eta}) \right] - \mathop{\mathbb{E}}_{q_{\boldsymbol{\phi}}} \left[ D_{\mathrm{KL}} \left( q_{\boldsymbol{\nu}} \,||\, p_{\boldsymbol{\psi}} \right) \right] - D_{\mathrm{KL}} \left( q_{\boldsymbol{\phi}} \,||\, p_{\boldsymbol{\theta}} \right) \equiv -\mathcal{L} \left( \boldsymbol{\theta}, \, \boldsymbol{\psi}, \, \boldsymbol{\phi}, \, \boldsymbol{\nu} \right), \tag{7}$$

---

1. Note, as we will see later, the prototype is arbitrary. I.e., the angle could take any value.

# Extended Abstract Track

where $q_{\boldsymbol{\nu}}(\boldsymbol{\eta}\,|\,\mathbf{x})$ and $q_{\boldsymbol{\phi}}(\hat{\mathbf{x}}\,|\,\mathbf{x})$ are NNs with parameters $\boldsymbol{\nu}$ and $\boldsymbol{\phi}$. By choosing $q_{\boldsymbol{\phi}}(\hat{\mathbf{x}}\,|\,\mathbf{x})$ to be a function invariant under $\mathcal{T}_{\boldsymbol{\eta}}$, we encourage $\hat{\mathbf{x}}$ not to contain any information about $\boldsymbol{\eta}$. Perhaps counter-intuitively, and in contrast with previous approaches to learning invariances (van der Wilk et al., 2018; Benton et al., 2020; Schwöbel et al., 2021; van der Ouderaa and van der Wilk, 2022; Immer et al., 2022), we do not make this NN *partially* invariant. Nonetheless, as we will show, we can still learn a degree of invariance.

We now present three informal conjectures to explain some of the design choices for the model in Figure 1. We illustrate these conjectures by means of examples. In each example we will assume that $\mathcal{T}_{\eta}$ is counter-clockwise rotation, and thus $\eta$ is the angle of rotation.

**Conjecture 1: $p_{\boldsymbol{\psi}}(\eta\,|\,\hat{\mathbf{x}})$ must be sufficiently flexible.** Consider a dataset of slightly rotated '8's: $\{8, \text{\reflectbox{8}}, \text{\rotatebox{30}{8}}\}$. Let us assume that the prototype is '8'. Table 1 shows $p(\eta\,|\,\mathbf{x}, \hat{\mathbf{x}})$, the true distribution for $\eta$ given $\mathbf{x}$ and $\hat{\mathbf{x}}$, under the data generating process. Because '8' is symmetric, $p(\eta\,|\,\mathbf{x}, \hat{\mathbf{x}})$ is the sum of two deltas. Figure 2 compares the learned $p_{\boldsymbol{\psi}}(\eta\,|\,\hat{\mathbf{x}})$ given a *simple* uni-modal Gaussian family and a more *flexible* bi-modal mixture-of-Gaussian family with the aggregate true distribution $p(\eta\,|\,\hat{\mathbf{x}}) = \sum_{\mathbf{x} \in \{8, \text{\reflectbox{8}}, \text{\rotatebox{30}{8}}\}} p(\eta\,|\,\mathbf{x}, \hat{\mathbf{x}})^2$. Here, the simple distribution is not flexible enough, which results in a large amount of probability mass being wasted on angles with low density under the true data generating process. The definition of 'flexible enough' will, of course, vary depending on the problem.

Table 1: True distribution for $\eta$ given $\mathbf{x}$ and $\hat{\mathbf{x}}$.

| $\mathbf{x}$ | $\hat{\mathbf{x}}$ | $p(\eta\,|\,\mathbf{x}, \hat{\mathbf{x}})$ |
|---|---|---|
| 8 | 8 | $0.5 \cdot \delta(\eta - 0°) + 0.5 \cdot \delta(\eta - 180°)$ |
| $\text{\reflectbox{8}}$ | 8 | $0.5 \cdot \delta(\eta - 30°) + 0.5 \cdot \delta(\eta + 150°)$ |
| $\text{\rotatebox{30}{8}}$ | 8 | $0.5 \cdot \delta(\eta + 30°) + 0.5 \cdot \delta(\eta - 150°)$ |

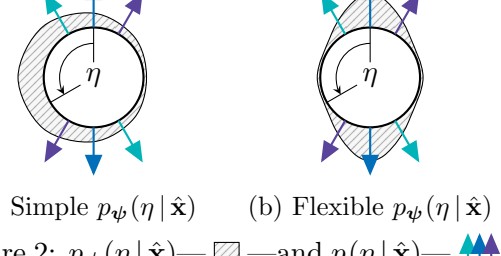

(a) Simple $p_{\boldsymbol{\psi}}(\eta\,|\,\hat{\mathbf{x}})$    (b) Flexible $p_{\boldsymbol{\psi}}(\eta\,|\,\hat{\mathbf{x}})$

Figure 2: $p_{\boldsymbol{\psi}}(\eta\,|\,\hat{\mathbf{x}})$—▨—and $p(\eta\,|\,\hat{\mathbf{x}})$—▲▲▲.

**Conjecture 2: $q_{\boldsymbol{\phi}}(\hat{\mathbf{x}}\,|\,\mathbf{x})$ must be *fully* invariant w.r.t $\eta$.** Consider a dataset of slightly rotated '2's: $\{2, \text{\reflectbox{2}}, \text{\rotatebox{30}{2}}\}$. Assume that the prototype is predicted by a *partially* invariant $q_{\boldsymbol{\phi}}(\hat{\mathbf{x}}\,|\,\mathbf{x})$. That is, $q_{\boldsymbol{\phi}}(\hat{\mathbf{x}}\,|\,\mathbf{x})$ is only similar for '2's which are equivalent under rotations smaller than $\pm\rho$. Table 2 shows the predicted prototypes and the corresponding distributions over $\eta$ for two cases: $\rho \geq 60°$ and $60° > \rho$. The threshold is $60°$ because each '2' can be transformed into each other '2' by a rotation of $\pm 60°$ or less. In the first case, each '2' maps to the same prototype. However, in the second case, each '2' can have a unique prototype. Thus, the prototype contains rotation information, which is invalid under our generative model of the data. Unfortunately, as illustrated by Figure 3, in the second case $p_{\boldsymbol{\psi}}(\eta\,|\,\hat{\mathbf{x}})$ can become arbitrarily large, which corresponds to a higher marginal likelihood

$$p(\mathbf{x}) = \iint p(\mathbf{x}\,|\,\eta, \hat{\mathbf{x}})\, p_{\boldsymbol{\psi}}(\eta\,|\,\hat{\mathbf{x}})\, p_{\boldsymbol{\theta}}(\hat{\mathbf{x}})\, d\eta\, d\hat{\mathbf{x}}, \tag{8}$$

---

2. Note that $p_{\boldsymbol{\psi}}(\eta\,|\,\hat{\mathbf{x}})$ is encouraged to cover all of the modes of $p(\eta\,|\,\hat{\mathbf{x}})$, under the assumption that $q_{\boldsymbol{\nu}}(\boldsymbol{\eta}\,|\,\mathbf{x})$ is accurate, due to the $D_{\mathrm{KL}}\left(q_{\boldsymbol{\nu}}\,||\,p_{\boldsymbol{\psi}}\right)$ term in the ELBO.

assuming that $p(\mathbf{x} \mid \eta, \hat{\mathbf{x}})$ and $p_{\boldsymbol{\theta}}(\hat{\mathbf{x}})$ are roughly the same in both cases[3]. By making $q_{\boldsymbol{\phi}}(\hat{\mathbf{x}} \mid \mathbf{x})$ fully invariant, the prototypes will be the same, and we avoid this degenerate optima.

Table 2: True distribution for $\eta$ given $\mathbf{x}$ and $\hat{\mathbf{x}}$.

| | | $\rho \geq 60°$ | | $60° > \rho$ | |
|---|---|---|---|---|---|
| $\mathbf{x}$ | $\hat{\mathbf{x}}$ | $p(\eta \mid \mathbf{x}, \hat{\mathbf{x}})$ | $\hat{\mathbf{x}}$ | $p(\eta \mid \mathbf{x}, \hat{\mathbf{x}})$ | |
| 2 | 2 | $\delta(\eta - 0°)$ | 2 | $\delta(\eta - 0°)$ | |
| 2 | 2 | $\delta(\eta - 30°)$ | 2 | $\delta(\eta - 0°)$ | |
| 2 | 2 | $\delta(\eta + 30°)$ | 2 | $\delta(\eta - 0°)$ | |

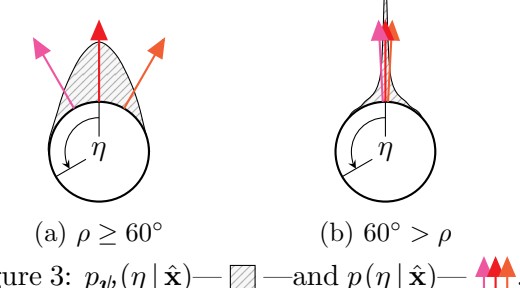

(a) $\rho \geq 60°$     (b) $60° > \rho$

Figure 3: $p_{\boldsymbol{\psi}}(\eta \mid \hat{\mathbf{x}})$— ▨ —and $p(\eta \mid \hat{\mathbf{x}})$— ▲▲▲.

**Conjecture 3: The distribution over $\eta$ must depend on $\hat{\mathbf{x}}$.** Consider a dataset of slightly rotated '2's and '8's: $\{2, 2, 2, 8, 8, 8\}$, with prototypes '2' and '8'. Table 3 shows $p(\eta \mid \mathbf{x}, \hat{\mathbf{x}})$, the true distribution over $\eta$. Figure 4 compares learned distributions over $\eta$ and $\eta$ given $\hat{\mathbf{x}}$. Note that in the first case, we are able to sample invalid digits such as $\{2, 2, 2\}$, which did not appear in the dataset. However, when $\eta$ depends on $\hat{\mathbf{x}}$, this does not occur.

Table 3: True distribution for $\eta$ given $\mathbf{x}$ and $\hat{\mathbf{x}}$.

| $\mathbf{x}$ | $\hat{\mathbf{x}}$ | $p(\eta \mid \mathbf{x}, \hat{\mathbf{x}})$ |
|---|---|---|
| 2 | 2 | $\delta(\eta - 0°)$ |
| 2 | 2 | $\delta(\eta - 30°)$ |
| 2 | 2 | $\delta(\eta + 30°)$ |
| 8 | 8 | $0.5 \cdot \delta(\eta - 0°) + 0.5 \cdot \delta(\eta - 180°)$ |
| 8 | 8 | $0.5 \cdot \delta(\eta - 30°) + 0.5 \cdot \delta(\eta + 150°)$ |
| 8 | 8 | $0.5 \cdot \delta(\eta + 30°) + 0.5 \cdot \delta(\eta - 150°)$ |

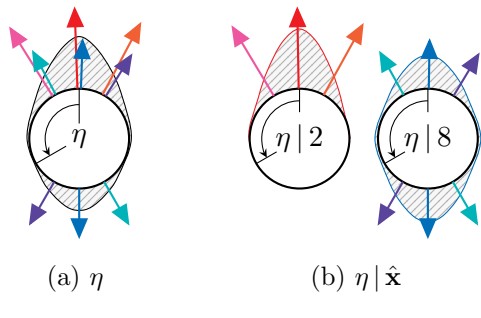

(a) $\eta$     (b) $\eta \mid \hat{\mathbf{x}}$

Figure 4: $p_{\boldsymbol{\psi}}(\eta)$ or $p_{\boldsymbol{\psi}}(\eta \mid \hat{\mathbf{x}})$— ▨ —and $p(\eta)$ or $p(\eta \mid \hat{\mathbf{x}})$— ▲▲▲▲.

## 3. Experiments

We validate that our model is able to learn invariance under symmetry transformations by training on a rotated version of the MNIST dataset. We choose $p_{\boldsymbol{\theta}}(\hat{\mathbf{x}})$ to be a VAE, and we implement $p_{\boldsymbol{\psi}}(\eta \mid \hat{\mathbf{x}})$ as a Neural Spline Flow (Durkan et al., 2019) $f_{\boldsymbol{\psi}}(\hat{\mathbf{x}})$ with base distribution $\mathcal{N}(\mu_{\boldsymbol{\psi}}(\hat{\mathbf{x}}), \sigma_{\boldsymbol{\psi}}(\hat{\mathbf{x}}))$ to ensure sufficient flexibility. $q_{\boldsymbol{\nu}}(\eta \mid \mathbf{x})$ is defined similarly. We do not place any restrictions on $\eta$, allowing it be learnt freely[4]. Further implementation details are in Appendix A. Figure 5 shows that our model successfully (i) learns prototype

---

3. In the first case of this simplified example, $p_{\boldsymbol{\theta}}(\hat{\mathbf{x}})$ could also become arbitrarily large. However, in practice since '2's vary in more ways than rotation—e.g., thickness, size, handwriting style—this will not occur.

4. Given prior knowledge of the appropriate bounds we could truncate the distributions over $\eta$ or encourage the distributions to place mass within the bounds with additional regularisation terms.

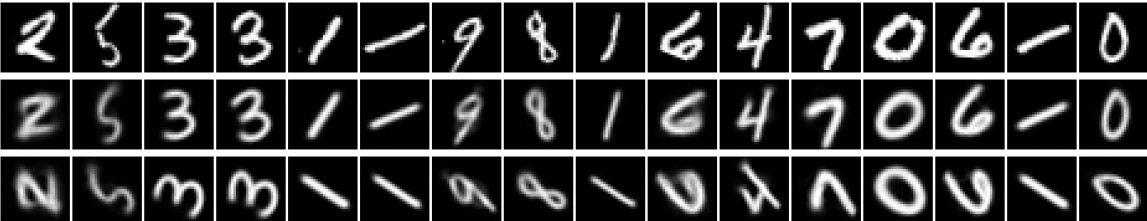

Figure 5: Learning invariance to rotations, $\eta \sim \mathcal{U}(-45°, 45°)$, of the MNIST dataset. **Top:** Samples $\mathbf{x}$ from the test set. **Mid:** Reconstructions of $\mathbf{x}$. **Bot:** prototypes $\hat{\mathbf{x}}$ for each $\mathbf{x}$.

images that are invariant to rotations—each $\hat{\mathbf{x}}$ is at the same arbitrary angle $\approx 135°$ regardless of the orientation of $\mathbf{x}$—and (ii) reconstructs rotated digits in the correct orientation. Figure 6 shows that the model behaves sensibly in the presence of different transformation intensities—as the digits are rotated more, the model predicts a wider range of angles. Note that as the degree of rotation in the data increases, so does the range of angles predicted by our model. This is in contrast to methods such as Benton et al. (2020) which learn a 'maximum data augmentation' which decreases as the data is transformed to a greater extent.

## 4. Conclusion

We have presented a method for learning a generative model with representations that are invariant to symmetry transformations. Our method learns these invariances in an unsupervised manner, without signal from an auxiliary task such as classification. We discussed various pitfalls for invariant generative models, which motivated our design choices. We also provided initial qualitative results, showing that the model can learn invariant representations for rotated MNIST digits and behaves reasonably under varying amounts of rotation.

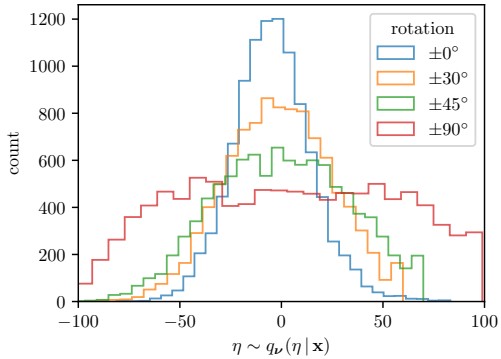

Figure 6: Samples from $q_{\boldsymbol{\nu}}(\eta \,|\, \mathbf{x})$ for digits rotated by $\eta \sim \mathcal{U}(\eta_{\min}, \eta_{\max})$.

However, this is all a first step towards learning invariant generative models. Important next steps include (i) formalization of our motivating conjectures, (ii) quantitative results with comparisons to baseline methods, (iii) extension to a wider range of transformations (e.g., other affine transformations, colour transformations), (iv) ablation studies for all of our design choices, (v) scaling up to more interesting problems than MNIST digits, and (vi) investigating our motivating example of improved data efficiency for invariant generative models.

Extended Abstract Track

## Acknowledgments

JUA acknowledges funding from the EPSRC, the Michael E. Fisher Studentship in Machine Learning, and the Qualcomm Innovation Fellowship. JA acknowledges support from Microsoft Research, through its PhD Scholarship Programme, and from the EPSRC. SP acknowledges support from the Harding Distinguished Postgraduate Scholars Programme Leverage Scheme. JMHL acknowledges support from a Turing AI Fellowship under grant EP/V023756/1.

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

# Appendix A. Additional Experimental Details

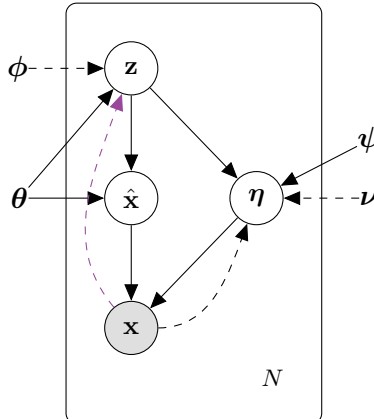

Figure 7: Generative (solid lines) and inference (dashed lines) models. Purple lines represent invariant NNs.

The full generative model for our experiments, modelling $p_{\boldsymbol{\theta}}(\hat{\mathbf{x}})$ with a VAE, is:

$$\mathbf{z} \sim p_{\boldsymbol{\theta}}(\mathbf{z}) = \mathcal{N}\left(\boldsymbol{\mu}_{\mathbf{z}},\, \boldsymbol{\sigma}_{\mathbf{z}} \cdot \mathbb{I}\right), \tag{9}$$

$$\hat{\mathbf{x}} \sim p_{\boldsymbol{\theta}}(\hat{\mathbf{x}} \,|\, \mathbf{z}), \tag{10}$$

$$\boldsymbol{\eta} \sim p_{\boldsymbol{\psi}}(\boldsymbol{\eta} \,|\, \mathbf{z}), \tag{11}$$

$$\mathbf{x} \sim p(\mathbf{x} \,|\, \hat{\mathbf{x}},\, \boldsymbol{\eta}). \tag{12}$$

The full graphical model is shown in Figure 7. We use the following ELBO for jointly learning the generative parameters $\{\boldsymbol{\theta},\, \boldsymbol{\psi}\}$ and the variational parameters $\{\boldsymbol{\phi},\, \boldsymbol{\nu}\}$:

$$\log p(\mathbf{x}) = \log \iiint p(\mathbf{x},\, \hat{\mathbf{x}},\, \mathbf{z},\, \boldsymbol{\eta})\, d\hat{\mathbf{x}}\, d\mathbf{z}\, d\boldsymbol{\eta} \tag{13}$$

$$= \log \iiint p(\mathbf{x} \,|\, \hat{\mathbf{x}},\, \boldsymbol{\eta})\, p_{\boldsymbol{\theta}}(\hat{\mathbf{x}} \,|\, \mathbf{z})\, p_{\boldsymbol{\psi}}(\boldsymbol{\eta} \,|\, \mathbf{z})\, p_{\boldsymbol{\theta}}(\mathbf{z})\, d\hat{\mathbf{x}}\, d\boldsymbol{\eta}\, d\mathbf{z} \tag{14}$$

$$= \log \iiint p(\mathbf{x} \,|\, \hat{\mathbf{x}},\, \boldsymbol{\eta})\, p_{\boldsymbol{\theta}}(\hat{\mathbf{x}} \,|\, \mathbf{z})\, p_{\boldsymbol{\psi}}(\boldsymbol{\eta} \,|\, \mathbf{z})\, p_{\boldsymbol{\theta}}(\mathbf{z})\, \frac{q_{\boldsymbol{\nu}}(\boldsymbol{\eta} \,|\, \mathbf{x})}{q_{\boldsymbol{\nu}}(\boldsymbol{\eta} \,|\, \mathbf{x})} \frac{q_{\boldsymbol{\phi}}(\mathbf{z} \,|\, \mathbf{x})}{q_{\boldsymbol{\phi}}(\mathbf{z} \,|\, \mathbf{x})}\, d\hat{\mathbf{x}}\, d\boldsymbol{\eta}\, d\mathbf{z} \tag{15}$$

$$= \log \mathop{\mathbb{E}}_{q_{\boldsymbol{\nu}}(\boldsymbol{\eta} \,|\, \mathbf{x})\, q_{\boldsymbol{\phi}}(\mathbf{z} \,|\, \mathbf{x})\, p_{\boldsymbol{\theta}}(\hat{\mathbf{x}} \,|\, \mathbf{z})} \left[ \frac{\mathcal{N}(\mathbf{x} \,|\, \mathcal{T}_{\boldsymbol{\eta}}(\hat{\mathbf{x}}),\, \sigma\mathbb{I})\, p_{\boldsymbol{\psi}}(\boldsymbol{\eta} \,|\, \mathbf{z})\, p_{\boldsymbol{\theta}}(\mathbf{z})}{q_{\boldsymbol{\nu}}(\boldsymbol{\eta} \,|\, \mathbf{x})\, q_{\boldsymbol{\phi}}(\mathbf{z} \,|\, \mathbf{x})} \right] \tag{16}$$

$$\geq \mathop{\mathbb{E}}_{q_{\boldsymbol{\nu}}(\boldsymbol{\eta} \,|\, \mathbf{x})\, q_{\boldsymbol{\phi}}(\mathbf{z} \,|\, \mathbf{x})\, p_{\boldsymbol{\theta}}(\hat{\mathbf{x}} \,|\, \mathbf{z})} \left[ \log \mathcal{N}(\mathbf{x} \,|\, \mathcal{T}_{\boldsymbol{\eta}}(\hat{\mathbf{x}}),\, \sigma\mathbb{I}) + \log p_{\boldsymbol{\psi}}(\boldsymbol{\eta} \,|\, \mathbf{z}) + \log p_{\boldsymbol{\theta}}(\mathbf{z}) - \log q_{\boldsymbol{\nu}}(\boldsymbol{\eta} \,|\, \mathbf{x}) - \log q_{\boldsymbol{\phi}}(\mathbf{z} \,|\, \mathbf{x}) \right] \tag{17}$$

$$\equiv \mathcal{L}\left(\boldsymbol{\theta},\, \boldsymbol{\psi},\, \boldsymbol{\phi},\, \boldsymbol{\nu}\right). \tag{18}$$

We estimate the ELBO with a single Monte-Carlo sample. We follow Benton et al. (2020); van der Ouderaa and van der Wilk (2022); Immer et al. (2022) in constructing approximately invariant NNs via expectation. Specifically, we construct an invariant encoder $q_{\boldsymbol{\phi}}(\mathbf{z} \,|\, \mathbf{x})$ from a non-invariant encoder $\hat{q}_{\boldsymbol{\phi}}(\mathbf{z} \,|\, \mathbf{x})$:

$$q_{\boldsymbol{\phi}}(\mathbf{z} \,|\, \mathbf{x}) \equiv \mathbb{E}_{\boldsymbol{\eta}}\left[\hat{q}_{\boldsymbol{\phi}}(\mathbf{z} \,|\, \mathcal{T}_{\boldsymbol{\eta}}(\mathbf{x}))\right]. \tag{19}$$

# Extended Abstract Track

We approximate this expectation using 10 Monte-Carlo samples.

We use `jax` with `flax`, `optax`, and `distrax` to implement our model. Our VAE encoder $q_\phi(\mathbf{z} \mid \mathbf{x})$ is a 3-layer CNN, with $\{64, 128, 256\}$ channels parameterising a heteroskedastic Gaussian distribution. The VAE decoder $p_\theta(\hat{\mathbf{x}} \mid \mathbf{z})$ is a 3-layer CNN, with transposed convolutions of $\{256, 128, 64\}$ channels, parameterising a homoskedastic Gaussian distribution. The latent dimension is 20. $p_\psi(\boldsymbol{\eta} \mid \mathbf{z})$ is a Neural Spline Flow with four bins, two flow layers, and base distribution $\mathcal{N}(\mu_\psi(\mathbf{z}), \sigma_\psi(\mathbf{z}))$ where the $\mu_\psi$ and $\sigma_\psi$ are a two-layer MLPs with hidden sizes $\{64, 32\}$. $q_\nu(\boldsymbol{\eta} \mid \mathbf{x})$ follows the same structure. For both flow models, each flow layer is parameterised by a 2-layer MLP of with hidden sizes $\{64, 32\}$.

All neural networks use `relu` activation functions. The VAE encoder and decoder use `LayerNorm`. We use the `adamw` optimizer with a initial learning rate of $1 \times 10^{-4}$ and a weight decay of $1 \times 10^{-4}$. We train for 15000 steps with a batch size of 256. We linearly increase the LR to $1 \times 10^{-3}$ over the first 500 training steps and then use a cosine schedule to reduce it back to $1 \times 10^{-4}$ by the end of training. We multiply the VAE KL-divergence by a factor $\beta$, which has an initial value of 10 and is decayed to 1, using a cosine schedule, by the end of training.

