# OpenReview forum: "Learning Generative Models with Invariance to Symmetries"
_NeurIPS.cc/2022/Workshop/NeurReps — NeurReps 2022 Poster_

### Official Review · Reviewer_oTiA · 2022-10-05
**Review for Learning Generative Models with Invariance to Symmetries**

**Confidence:** 5
**Soundness:** 3
**Presentation:** 4
**Contribution:** 3
**Overall Rating:** 7

**Summary:**

The paper proposes a probabilistic framework relying on latent variables to learn invariant generative models where the invariance is unknown to the model. A variational lower bound objective is derived and empirically the authors perform experiments on a partially rotated version of MNIST to demonstrate their approach.

**Questions:**

My main question is for the experiments, do you still not need to know the support of the distributions apriori? For example how is the range of $\eta$ restricted? What happens in the case of multiple invariants not just rotation? Consider the group $E(2)$ which has multiple symmetries and can also be applied to certain image datasets. How would you elevate your current modeling framework to accommodate this?

**Limitations:**

The developed theory lacks rigor.

**Recommended Decision:**

3: Accept

**Relevance:**

3: Solid fit

**Strengths And Weaknesses:**

The paper is very clearly written and from an idea perspective, it is very clear and natural. The proposed graphical model is also sensible and standardizing pose has been previously done in the discriminative literature. Overall this paper is a nice idea with early promise based on preliminary experiments. As for weaknesses the main weakness is that the three propositions are not really propositions and proof by example is not really a proof. I encourage the authors to make these statements a bit more formal and crisper.

**Submission Track:**

Extended Abstract (4 Page)

---

### Official Review · Reviewer_PYoK · 2022-10-10

**Confidence:** 4
**Soundness:** 2
**Presentation:** 2
**Contribution:** 2
**Overall Rating:** 5

**Summary:**

The authors propose a framework to learn a joint distribution over data and parameters of certain transformations to which the data is invariant. They apply the developed model to handwritten digits and demonstrate that the framework is able to capture the invariance of the images to planar rotations.

**Questions:**

The authors provide an extensive list of suggestions in the paper's conclusion. Implementing any of them would strengthen the paper considerably.

- My suggestion would be to analyze boundary cases. It is interesting how the model handles "6" and rotated 180° "9", i.e. what is the probability distribution over $\eta$  given any of those?
- Also, how the model reacts to noise in data and/or transformation parameters $\eta$?
- In the second section, authors write: "by choosing $q_\phi (\hat x | x )$ to be invariant to $\eta$, we encourage $x$ not to contain any information about the symmetry for $T_\eta$. My question is: how do you prove invariance, i.e. $q_\phi (\hat x | x ) = q_\phi (T_\eta(\hat x) | x )$? An additional experiment demonstrating that empirically would be a nice addition, e.g. $| log q_\phi (\hat x | x ) - log  q_\phi (T_\eta(\hat x) | x ) |$ vs $\eta$.



**Limitations:**

Limitations are not listed, but directions for further work are given in the conclusion.

**Recommended Decision:**

2: Borderline

**Relevance:**

3: Solid fit

**Strengths And Weaknesses:**

**Strengths**
- the paper addresses an important problem with a novel approach.
- the results are sound enough. Particularly, generated prototypes in Fig. 6 look convincing.
- authors discuss design choices thoroughly and provide vivid examples to support them.

**Weaknesses**
- the paper is not clearly written. There are abbreviations that are used but not defined (see NN and r.v. in the introduction, ELBO in the 2nd section). "learning the invariance" is not defined.
- I find the abstract confusing. The authors say: "most methods require ... prior knowledge of the symmetries" and claim that their goal is to "learn a generative model of handwritten digits that
is invariant to rotation". I would assume that the model would not be aware of symmetry transformations, but then the authors define $p(x |\hat x, \eta) = N(T_\eta(\hat x), I)$. Therefore, they seem to actually implement the symmetry as inductive bias into the model as rotation symmetry is encoded into the model.
- typo in the introduction: "One class of inductive bias that *improves does improve* efficiency..."
- the authors consistently write "invariance to symmetries", which is technically incorrect. The better wording would be "invariance under/to a symmetry transformation".
- figures from section 2 look good but, combined with tables, make the section appear tangled. They probably would fit better into the Appendix.
- no comparison against baseline approaches.


**Submission Track:**

Extended Abstract (4 Page)

---

### Official Review · Reviewer_3u4G · 2022-10-15
**Invariant Generative Models**

**Confidence:** 3
**Soundness:** 3
**Presentation:** 4
**Contribution:** 3
**Overall Rating:** 7

**Summary:**

This extended abstract presents the idea of invariant learning in generative models for a given transformation. The latent variable does not contain any information about this transformation but the model is able to reconstruct the data in the correct orientation. The authors give some intuition about the proposed design choices in the generative framework and validate their claim through some prototype experiments on the rotated MNIST dataset.

**Questions:**

1. Regarding Proposition 3 in the paper, it is an odd choice to have $\eta$ depending on $\hat{x}$, considering the idea is to learn the symmetry and for this all possible cases of $\theta$ should be considered. For example, with digit 8, the model should be able to learn the additional symmetry or recognize it during the generation process, and for digit two allow all the possible valid rotations. Could you explain this design choice?

2. Were experiments performed at different $\hat{x}$? Did they give similar results?

3. What is the scope of this idea for making fully invariant and equivariant generative models? And how do you think they compare with previous works like VAEs with equivariant priors, etc?

**Limitations:**

See Weakness

**Recommended Decision:**

3: Accept

**Relevance:**

3: Solid fit

**Strengths And Weaknesses:**

The paper presents an interesting idea regarding learning symmetries in generative models without data augmentation. The paper constructs an invariant encoder from a noninvariant encoder with invariant latent variables. The prototype experiments validate the idea and contribute to making generative models efficient. This work can be potential used for learning partial or approximate symmetries.

Although, there the idea of learning the degree of invariance with a self-symmetric object/digit is seemingly odd as that can lead to a biased probability mass distribution over angles.

**Submission Track:**

Extended Abstract (4 Page)

---

### Decision · Program_Chairs · 2022-10-21

Accept (Poster)